# Optimal Generative Cyclic Transport between Image and Text

## Abstract

Deep generative models, such as vision-language models (VLMs) and diffusion models (DMs), have achieved remarkable success in cross-modality generation tasks. However, the cyclic transformation of text → image → text often fails to secure an exact match between the original and the reconstructed content. In this work, we attempt to address this challenge by utilizing a deterministic function to guide the reconstruction of precise information via generative models. Using a color histogram as guidance, we first identify a soft prompt to generate the desired text using a language model and map the soft prompt to a target histogram. We then utilize the target color histogram as a constraint for the diffusion model and formulate the intervention as an optimal transport problem. As a result, the generated image has the exact color histogram as the target, which can be converted to a soft prompt deterministically for reconstructing the text. This allows the generated images to entail arbitrary forms of text (e.g., natural text, code, URLs, etc.) while ensuring the visual content is as natural as possible. Our method offers significant potential for applications on histogram-constrained generation, such as steganography and conditional generation in latent space with semantic meanings.

## 1 Introduction

While deep generative models, such as vision-language models (VLMs) (Radford et al., 2021; Liu et al., 2024a;b) and diffusion models (DMs) (Sohl-Dickstein et al., 2015; Ho et al., 2020; Dhariwal & Nichol, 2021; Rombach et al., 2022; Esser et al., 2024), have achieved significant success in cross-modality generation tasks, challenges persist in preserving exact information during cyclic transformations, such as text → image → text. A primary cause of this difficulty lies in the inherent ambiguities and information loss that occur when converting between modalities. As illustrated by the Diffusion generation process in Fig. 1, when generating images from textual descriptions, even detailed prompts may lack sufficient specificity to fully capture all nuances of the original text (i.e., the generated image only exhibits one possible depiction among many, reducing complex semantic information into abstract or less granular forms).

Similarly, when converting images back into text (i.e., the VLM captioning process in Fig. 1), the process introduces further complexity, as the visual data can potentially be captioned in a variety of ways, where multiple synonymous terms or paraphrases could be adopted. This multiplicity of valid textual interpretations contributes to information drift, as subtle word choices or phrasing variations lead to the loss of precise meanings or intended content. Consequently, during cross-domain transformations, each stage introduces opportunities for information to be diluted, approximated, or even transformed into concepts that, while similar, fail to match the original input exactly, particularly when reconstructing structured data or metadata embedded within the content. Such information loss is fatal to applications that rely on accurate cyclic transformations between multiple modalities, leading to sub-optimal solutions and multi-modal representations that are not specific enough.

In this work, we propose using a vector as the medium so that the diffusion model can generate an image where such a vector can be derived deterministically. On top of that, a large language model (LLM) can decode such a vector into exact information. Hence, we can achieve the cyclic generation with no loss. As demonstrated in Fig. 1, we can precisely decode the generation configurations of the image, which can later be used to reproduce the entire generation setup. Some prior work has explored a systematic approach of baking QR code into images (Wu et al., 2024), where at most 4K

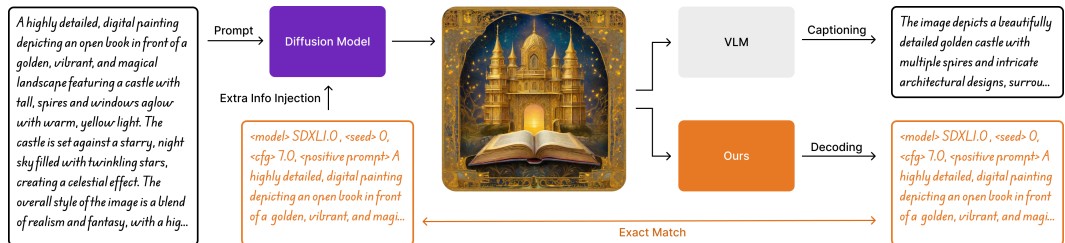

Figure 1: Demonstration of exact text decoding of OGCT compared to VLM captioning.

alphanumeric characters can be encoded in the ideal case. In contrast, by using the medium vector as the constraint, thousands of exact text tokens can be decoded from this single vector, where 1) all Unicode characters can be encoded, and 2) there is almost no impact on the image content.

To achieve cyclic generation and exact information decoding, several key properties are essential to the design of the medium vector: First, it should be compact, allowing efficient storage and processing while maintaining the integrity of the encoded information. Second, the vector shall be robust to minor perturbations, ensuring reliable decoding performance. Third, it should support deterministic computations, eliminating randomness in the encoding-decoding cycle. Hence, histograms in pixel or latent space emerge as a viable choice for implementation as they satisfy all the aforementioned properties and offer sufficient flexibility for image generation.

Accordingly, the cyclic transformation problem was decoupled into two subproblems. During the diffusion process, the problem lies in how to enforce the generated image to have a desired histogram (Sec. 2.4, Sec. C). Despite the success of ControlNet (Zhang et al., 2023) and LoRAs (Hu et al., 2022) in controllable image generation, we showed that their controls over the color histogram cannot ensure exact match (Sec. A). Instead, directly manipulating the feature map seems a more reliable approach, where the histogram matching can be formulated as an optimal transport problem. Based on the optimal transport plan, we can transform the image to follow the exact target histogram.

For the decoding side, the problem now becomes finding an embedding to reconstruct the encoded text, where soft prompt tuning (Lester et al., 2021) offers an efficient and elegant solution. By tuning a single pre-pended trainable token, we can obtain a medium vector (i.e., the soft prompt itself) that can prompt an LLM to generate desirable text, which can be baked into the diffusion model seamlessly for exact histogram encoding.

We envision that a variety of applications enabled by the proposed framework. For example, one can encode proprietary information (e.g. copyright notices and licensing terms) directly into their work or hide other sensitive information for secure transmission. The appearance of the encoded image is natural to human viewers, while only those with access to the secured generative model decoder could decode the hidden information. Moreover, apart from being a steganography technique, it is also widely applicable for histogram-constrained diffusion generation, where the histogram could correspond to arbitrary histograms or latent spaces with semantic meanings.

## 2 METHOD

### 2.1 PRELIMINARIES

**Vision-Language Models.** Motivated by the growing need for systems capable of understanding and generating content across both modalities, Vision-Language Models (VLMs) have been proposed to connect the domain of two critical modalities – text and image, empowering numerous real-world tasks involving the interplay between images and text, such as captioning, visual question answering, and cross-modal retrieval. Early approaches typically relied on separate models for vision and language tasks, often using feature fusion techniques. However, these approaches struggled with generalization across diverse tasks. To address this, researchers have developed large pre-trained VLMs that leverage massive datasets of paired image-text data, enabling the models to learn richer joint representations of both modalities (e.g., CLIP (Radford et al., 2021), ALIGN (Jia et al., 2021a)). Recently, SOTA models such as the LLaVA (Large Language and Vision Assistant) family (Liu et al., 2024a;b) have further advanced the field by combining vision-language

pre-training with large-scale language models (LLMs) (Dubey et al., 2024), allowing for more accurate and versatile cross-modal reasoning and generation. These models are increasingly capable of generalizing across a wide range of tasks. However, challenges remain in ensuring precision and specificity in tasks requiring exact cross-modal correspondence since the mapping between text and image is arguably not a bijection up to the ambiguities of synonyms and abstract entities.

**Diffusion Models.** Diffusion Models (DMs) (Sohl-Dickstein et al., 2015; Ho et al., 2020; Dhariwal & Nichol, 2021; Rombach et al., 2022) have emerged as the predominant method for image generation, surpassing Generative Adversarial Networks (GANs) (Goodfellow et al., 2014; Zhu et al., 2017) due to their stability in training and ability to generate high-quality, diverse samples. Unlike GANs, which rely on adversarial training between a generator and discriminator, DMs learn to generate images by modeling a stochastic process that gradually transforms noise into coherent images. This process is typically governed by a forward diffusion process, which adds Gaussian noise to an image over time, and a reverse process, which removes noise step by step to recover the data distribution. Mathematically, the forward process can be defined as:

$$q(\mathbf{x}_t|\mathbf{x}_{t-1}) = \mathcal{N}(\mathbf{x}_t; \sqrt{\alpha_t}\mathbf{x}_{t-1}, (1 - \alpha_t)\mathbf{I}), \tag{1}$$

which essentially characterizes a Gaussian distribution where noise is progressively added to the data $\mathbf{x}_0$ based on certain scheduling $\{\alpha_t\}_{t=1:T}$ as the time step $t$ increases. Moreover, the reverse process is governed by the following ODE:

$$d\mathbf{x} = \left(f(\mathbf{x}, t) - g^2(t)\nabla_{\mathbf{x}} \log p_t(\mathbf{x})\right) dt + g(t)d\mathbf{w}, \tag{2}$$

where $f(\mathbf{x}, t)$ denotes the drift term, $g(t)$ denotes the diffusion coefficient, and $\nabla_{\mathbf{x}} \log p_t(\mathbf{x})$ is the score function that helps guide the denoising process.

Notably, Diffusion Models are particularly compelling in conditional generation tasks, where the image generation is guided by external information. Condition-specific embeddings guide the diffusion process toward generating images that match the given condition input. This has led to the success of models like DALL·E (Ramesh et al., 2022), Stable Diffusion (Rombach et al., 2022), and most recently Flux.1 (BlackForestLabs, 2024), where textual prompt is arguably the most popular condition modality. While flexible, the text-guided diffusion does not necessarily secure the exact match of text reconstruction when captioning diffusion-generated images to retrieve the textual prompts (used for generation) via VLMs. This motivates us to look into the potential solutions that secure exact reconstruction during cyclic cross-modality transformations.

## 2.2 METHOD OVERVIEW

Fig. 2 manifests the workflow of the proposed Optimal Generative Cyclic Transport (OGCT) framework. We first optimize towards a soft prompt embedding (i.e., a single trainable token) that will condition the LLM to generate the encoded text. Then, we transform the soft prompt embedding into a histogram vector (non-negative, sum to 1) with a deterministic mapping. By intervening in the text-to-image generation process, we can enforce the generated image to have the color histogram matching the pre-computed histogram vector. As a result, when someone decodes the image (i.e., calculates the color histogram and transforms it back to the embedding form), the reconstructed soft prompt will condition the LLM to reconstruct the encoded text, which can be as long as hundreds to thousands of tokens. We introduce the three key modules of our framework in the following subsections. Sec. 2.3 covers the optimization of the ideal soft prompt as well as the transformation between the soft prompt embedding and the histogram vector. Sec. 2.4 details how we intervene in the diffusion process while preserving the quality of the generated image. Sec. 2.5 formulates histogram matching as an optimal transport problem, where the binning strategy can be decoupled from the closeness of color values.

## 2.3 SOFT PROMPT OPTIMIZATION

Prompt tuning (Lester et al., 2021) is a parameter-efficient fine-tuning (PEFT) technique commonly employed in natural language processing (NLP) tasks to adapt large pre-trained language models (LLMs) to specific downstream tasks without updating the entire model. Rather than fine-tuning all of the model's parameters, prompt tuning involves learning a set of soft prompts or continuous embeddings that are prepended to the input text—that guide the language model's behavior. These soft

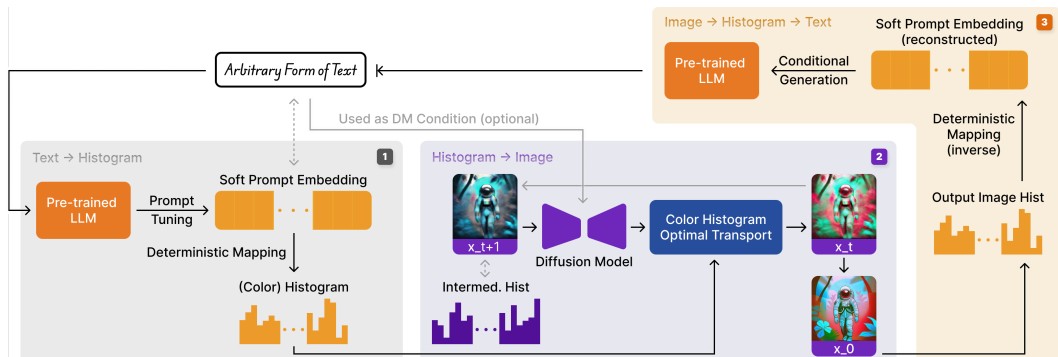

Figure 2: Overview of the Optimal Generative Cyclic Transport (OGCT) framework. OGCT enables the exact reconstruction of arbitrary forms of text (up to 1K tokens) by encoding the soft prompt embedding via the color histogram of the generated image. Zoom in for the best view.

prompts act as task-specific instructions that can condition the model to generate appropriate outputs for a given task, making the process more efficient in terms of both computation and memory.

Consequently, prompt tuning allows identifying a soft prompt that effectively conditions the LLM to output an exact sequence of tokens (i.e., any form of text). By optimizing the embeddings of the soft prompt, it becomes possible to precisely control the model's output, ensuring that the generated sequence matches any predefined target, such as natural text, code, URLs, or some mixture of them. Formally, aim to maximize the likelihood of the target text sequence $\mathbf{y} = (y_1, y_2, \cdots, y_T)$ given a learned soft prompt $\mathbf{p} \in \mathbb{R}^d$, where $d = 4096$ in practice. Given an LLM parameterized by $\Theta$, the training objective can be expressed as minimizing the conditional negative log-likelihood:

$$\mathcal{L}(\mathbf{p}) = \sum_{t=1}^{T} \log P(y_t | \mathbf{p}, y_{<t}; \Theta), \tag{3}$$

where $P(y_t | \mathbf{p}, y_{<t}; \Theta)$ models the probability of the token $y_t$ at time step $t$, conditioned on the soft prompt $\mathbf{p}$ and the previously generated tokens $y_{<t}$ and the sum is taken over the length $T$ of the target sequence. By optimizing the soft prompt $\mathbf{p}$ towards this objective, we can obtain a proper prior context in the embedding space that guides the model to produce the exact desired sequence.

However, a direct optimization with respect to the objective in Eq. 3 can be problematic, as we are taking gradient steps in the embedding space without any constraints. This can potentially lead to slow convergence as each gradient update would change both the magnitude and the direction of the soft prompt embedding. In contrast, we propose to rescale the embedding to some fixed norm so that the soft prompt can be dedicated to learning the optimal direction that conditions the LLM to generate the ideal output.

Noticing that the proposed approach inherently favors decoder-only models as opposed to encoder-decoder ones. On the one hand, decoder-only models are less constrained by the complex input-output alignments as in encoder-decoder models, where the input must be fully encoded before decoding begins; on the other hand, the optimization process for decoder-only models are simpler and more efficient, as the soft prompt embeddings can be seamlessly integrated into the beginning of the input sequence and each predicted token solely relies on the previous context (including the soft prompt itself) due to the auto-regressive nature.

After obtaining a proper soft prompt $\mathbf{p}$ that entails the target sequence $\mathbf{y}$, we can then convert it to some valid histogram $\mathbf{h} \in \mathbb{R}^d$ via a simple deterministic mapping. Formally, we define the embedding-to-histogram mapping as follows:

$$\mathbf{h} = f(\mathbf{p}) = \frac{\exp(\mathbf{p})}{\sum_i \exp(\mathbf{p}_i)}, \tag{4}$$

where the exponential function ensures the non-negativity of all entries and the normalization ensures all the entries of $\mathbf{h}$ sum to 1. As for the inverse mapping $f^{-1}$, we are essentially looking for some scaling factor $k \in \mathbb{R}$ such that $||\ln(k\mathbf{h})|| = ||\mathbf{p}||$. Given that the target embedding vector

has a pre-defined fixed norm, the value of $k$ can be efficiently solved. In this way, we can always find a unique solution that constructs the one-to-one mapping between $\mathbf{p}$ and $\mathbf{h}$, where no additional information is required during the decoding stage.

## 2.4 Histogram-Conditioned Diffusion Generation

After obtaining a valid color histogram $\mathbf{h}$, we wish to perturb the output of a diffusion model in a way that aligns the generated image with $\mathbf{h}$, while minimally affecting the overall image generation process. To achieve this, we perturb the endpoint of the diffusion process (i.e., $\mathbf{z}_0$) during inference. More specifically, we seek to apply this perturbation at a subset of inference time steps, adjusting the intermediate predictions of $\mathbf{z}_0$, and add the proper amount of noise back to the perturbed prediction $\mathbf{z}_0'$ to continue the inference procedure. Formally, at some intermediate time step $t$, we have:

$$\mathbf{z}_0 = \frac{1}{\sqrt{\bar{\alpha}_t}}\mathbf{z}_t - \frac{\sqrt{1-\bar{\alpha}_t}}{\sqrt{\bar{\alpha}_t}}\epsilon_\theta(\mathbf{z}_t, t), \tag{5}$$

where $\epsilon_\theta(\mathbf{z}_t, t)$ denotes the predicted noise and $\{\bar{\alpha}\}_{t=1:T}$ denote the cumulative noise factors. Then given some color histogram perturbation function $\varphi(\cdot)$ and the perturbed output image $\mathbf{z}_0' = \varphi(\mathbf{z}_0)$, we can calculate the updated signal on the diffusion trajectory as follows:

$$\mathbf{z}_{t-1} = \sqrt{\bar{\alpha}_{t-1}}\mathbf{z}_0' + \sqrt{1-\bar{\alpha}_{t-1}}\epsilon_\theta(\mathbf{z}_t, t). \tag{6}$$

In general, we apply the perturbation function $\varphi(\cdot)$ on the predictions of $\mathbf{z}_0$ at some intermediate time steps as well as the last step, and the diffusion model would correct its trajectory throughout the inference process. This ensures that the color histogram of the generated image conforms to the target distribution while maintaining the overall integrity of the generated content, resulting in a proper balance in between. We also note that most state-of-the-art diffusion models now operate in the latent space, whereas the perturbation of color histograms generally happens in the pixel space, so the encoding and decoding process of the variational auto-encoders (VAEs) (Kingma, 2013) have been entailed in the perturbation function $\varphi(\cdot)$ for simplicity.

## 2.5 Histogram Matching with Optimal Transport

Given a general approach to condition the diffusion process on color histogram, the next problem is how to design perturbation function $\varphi(\cdot)$ properly, so that the diffusion trajectory does not collapse to pure noise or low-quality outputs with weird coloring.

One natural idea is to formulate the histogram matching problem as an optimal transport (OT) problem. Formally, given the source histogram $\mathbf{h}^{src}$ and the target $\mathbf{h}^{tgt}$, where $\mathbf{h}^{src}, \mathbf{h}^{tgt} \in \mathbb{R}^d$, we want to solve the following optimization problem:

$$\gamma = \arg\min_\gamma \langle\gamma, \mathbf{M}\rangle_F, \quad \text{s.t. } \gamma\mathbf{1} = \mathbf{h}^{src}, \gamma^T\mathbf{1} = \mathbf{h}^{tgt}, \gamma \geq 0, \tag{7}$$

where $\gamma \in \mathbb{R}^{d\times d}$ denotes the optimal transport plan, $\mathbf{M} \in \mathbb{R}^{d\times d}$ denotes the cost matrix calculated from the pairwise L1 distance between normalized RGB tuples (i.e., the center of each color bin). To match the embedding dimension of $d = 4096$ for the soft prompt, we quantize the 8-bit RGB values to 4-bit and solve for the optimal transport plan $\gamma$ by the displacement interpolation via partial mass transport between radial basis functions (RBFs). However, this naive way of transporting the pixels can lead to sub-optimal results, where the output image suffers from a strong color hue jitter as illustrated in Fig. 3(b). This is because the target color histogram converted from the soft prompt tends to evenly divide the pixels into different RGB color bins. In contrast, most bins are generally empty for natural images.

Luckily, this problem can be alleviated by adjusting the binning strategy. Intuitively, the color histogram divides the pixels into a set of abstract bins based on certain criteria, where the binning strategy does not necessarily depend on the closeness of RGB values. For example, one mediated approach is to relax one of the RGB channels and apply binning by the closeness of the other two channels. As shown in Fig. 3(c), the color hue seems to be consistent when controlling the green and blue channels and relaxing the red channel. This approach can be further generalized to assign the pixels to bins by a pre-defined look-up table, such that the division of color comes from random shuffling instead of any distance metrics. In this way, the output image in Fig. 3(d) can be extremely similar to the input after recoloring by the target histogram.

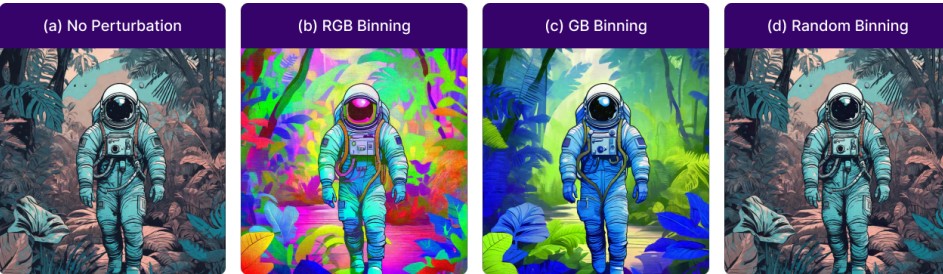

Figure 3: Qualitative comparison of different histogram binning strategies. Better view with color.

Noticing that we are still trying to tackle an optimal transport problem characterized by Eq. 7 but with parameters of different sizes. Formally, given $k$ random colors in each bin, we are effectively solving for $\gamma \in \mathbb{R}^{kd \times d}$ when $\mathbf{h}^{src} \in \mathbb{R}^{kd}, \mathbf{h}^{tgt} \in \mathbb{R}^d, \mathbf{M} \in \mathbb{R}^{kd \times d}$. For the $j$-th color in the $i$-th source bin, its closet color in the $p$-th target bin can be pre-computed and assigned to the cost matrix, such that:

$$\mathbf{M}_{ik+j,p} = \min_q ||\mathbf{c}_{ik+j}, \mathbf{c}_{pk+q}||_1, \quad \forall q \in [0, kd-1], q \in \mathbb{N}, \tag{8}$$

where $\mathbf{c} \in \mathbb{R}^{kd}$ corresponds to the flattened color book. In other words, we aim to find a more fine-grained level transport plan, not from bin to bin, but from color to color, under the constraint that the sum of the frequency of colors inside the target bins match the desired histogram. Note that we are free to choose the frequency of color inside every target bin as long as the sum matches. When applying the transport plan, we randomly sample a given number of pixels from each source color and set them to the closest color in the target bin. Since we use a random binning strategy, every color in the source bin will likely get transported to a similar color in the target bin, so the color change is barely noticeable.

## 2.6 OPTIMAL GENERATIVE CYCLIC TRANSPORT VIA COLOR HISTOGRAM

Having introduced the three key modules for performing OGCT with color histograms, we reiterate the workflow of the proposed method by the pseudo-code in Alg. 1. Overall, OGCT addresses the ambiguities involved in the process of cyclic cross-modality transformation (e.g., text $\to$ image $\to$ text), where more than one mapped targets can be identified as "correct" answers. The proposed framework adopts a perturbation function $\varphi(\cdot)$ to enforce the diffusion process endpoint (i.e., $\mathbf{z_0}$) to match some given target histogram, which can be converted to a soft prompt embedding for exact information reconstruction. We relax the constraints for the closeness of colors in our binning strategy, resulting in much better flexibility of image content without a loss of histogram precision. By a properly designed embedding-to-histogram mapping, one can decode the information from the image via the LLM without any extra information (e.g., normalizing factors). We also note that the color histogram based OGCT is essentially a super-set for the cyclic cross-modality transformation, as the encoded text can be decoupled from the text-to-image generation prompt and thus be of arbitrary form (e.g, natural text, code, URLs, etc.). More details can be found in Sec. 3.

## 3 EXPERIMENT

### 3.1 IMPLEMENTATION DETAILS

We set Llama-3.1 (Dubey et al., 2024) as the LLM, where each soft prompt embedding with $d = 4096$ dimensions. We use the AdamW (Loshchilov & Hutter, 2019) optimizer with an initial learning rate of 0.1, which decays by half every 200 training steps. We set the maximum number of training steps to 2000. We empirically rescale the soft prompt embedding to have a fixed L2 norm 40.0 after each gradient update. We generate images using the StableDiffusion-XL (Podell et al., 2023) and use the default resolution of $1024 \times 1024$. We use the DDIM (Song et al., 2020) noise scheduler with 50 inference steps. We apply histogram matching perturbation every 10 inference steps when the binning strategy depends on the closeness in the color space (e.g., RGB, RG); we only perturb the output image once in the case of random binning, as the perturbed image is close to unchanged. We

---

**Algorithm 1** Optimal Generative Cyclic Transport via Color Histogram

---

**Input:** Decoder-only LLM $\Theta_1$, diffusion model $\Theta_2$, target text sequence $\mathbf{y}$, learning rate scheduler $\eta(t)$, fixed norm $n$, textual prompt $\mathcal{P}$, perturbation time steps $\mathcal{T}$, color book $\mathbf{c}$.
**Output:** Output image $\mathcal{I}$, decoded text $\hat{\mathbf{y}}$

1: Initialize soft prompt embedding $\mathbf{p}$
2: **while** $\mathcal{L}(\mathbf{p}, \mathbf{y}; \Theta_1) >= \tau$ **do**                                        ▷ Prompt tuning loop
3:     $\mathbf{p} = \mathbf{p} - \eta(t) \cdot \nabla\mathcal{L}(\mathbf{p}, \mathbf{y}; \Theta_1)$          ▷ Soft prompt update with learning rate scheduler
4:     $\mathbf{p} = \texttt{norm-rescale}(\mathbf{p}, n)$                                  ▷ Rescale the soft prompt to fixed norm
5: **end while**
6: $\mathbf{h}^{tgt} = \exp(\mathbf{p})/(\sum_i \exp(\mathbf{p}_i))$                               ▷ Set target histogram
7: $\mathbf{z}_T = \texttt{Gaussian-sampling}()$                                   ▷ Initial noise sampling
8: **for** t = T:1 **do**                                                           ▷ Text-to-image diffusion loop
9:     $\mathbf{z}_0^t = \frac{1}{\sqrt{\bar{\alpha}_t}}\mathbf{z}_t - \frac{\sqrt{1-\bar{\alpha}_t}}{\sqrt{\bar{\alpha}_t}}\epsilon_\theta(\mathbf{z}_t, t; \mathcal{P}; \Theta_2)$       ▷ Trajectory endpoint prediction
10:     **if** $t \in \mathcal{T}$ **then**                                                ▷ Time step for perturbation
11:         $\mathbf{z}_0^{t'} = \texttt{OT-histogram-matching}(\mathbf{z}_0^t, \mathbf{h}^{tgt})$
12:         $\mathbf{z}_{t-1} = \sqrt{\bar{\alpha}_{t-1}}\mathbf{z}_0^{t'} + \sqrt{1 - \bar{\alpha}_{t-1}}\epsilon_\theta(\mathbf{z}_t, t; \mathcal{P}; \Theta_2)$        ▷ Perturbed update
13:     **else**
14:         $\mathbf{z}_{t-1} = \sqrt{\bar{\alpha}_{t-1}}\mathbf{z}_0^t + \sqrt{1 - \bar{\alpha}_{t-1}}\epsilon_\theta(\mathbf{z}_t, t; \mathcal{P}; \Theta_2)$        ▷ Regular update
15:     **end if**
16: **end for**
17: $\mathcal{I} = \texttt{VAE-decode}(\mathbf{z}_0)$                                           ▷ Output image
18: $\mathbf{h}^{recon}, \mathbf{p}^{recon} = \texttt{calculate-histogram}(\mathcal{I}; \mathbf{c})$        ▷ Soft prompt reconstruction
19: $\hat{\mathbf{y}} = \texttt{LLM-forward}(\mathbf{p}^{recon}; \Theta_1)$                           ▷ Information decoding
20: **return** $\mathcal{I}, \hat{\mathbf{y}}$

---

use a pre-defined color book of size 65536 (i.e., 16 colors per bin) to pre-compute the cost matrix for random binning. All the training and inference are performed on a single NVIDIA A100 GPU.

## 3.2 EVALUATION OF OGCT

**Qualitative Results.** To evaluate the information encoding capability of OGCT, we collect the README Markdown files from public GitHub [1] repositories with top-100 star numbers, which is essentially a mixture of multilingual natural text, code, and URLs. We randomly crop the raw text to have fixed number of tokens within $\{32, 64, 128, 256, 512\}$, resulting in 300 independent data samples at each token length. We use the prompt tuning and image generation configuration as specified in Sec. 3.1 by default. We collect and filter 148 textual prompts from Civitai [2] for image generation, which is randomly paired with a trained soft prompt during OGCT. Some qualitative results are presented in Fig. 4. From the figure we can observe that though relaxing one of the RGB channels in color binning helps reduce the variety of colors in the images, the controlled color channels may still suffer from weird coloring effect in some cases, which should be caused by the perturbation of color histogram. Notably, some of the content deviates from the unperturbed image, which demonstrates how diffusion model attempts to naturalize the image after matching to some target histograms. In comparison, random binning consistently gives output images that are extremely similar to the image without any perturbation. This demonstrates the superiority of decoupling the binning strategy from the closeness of colors, resulting in a flexible choice of colors while transporting a source color to some fixed target bin.

**Quantitative Results.** The quantitative results are presented in Tab. 1. In the prompt tuning stage, we report the success rate of finding a soft prompt that leads to the exact encoded text from the LLM as well as the training time. From the statistics, we can see that the optimization of a medium-length text sequence (i.e., less than 256 tokens) generally takes less than 20 seconds on a single NVIDIA A100 GPU with >99.7% success rate. We also note that it is empirically possible to find a 4096-dimensional soft prompt that generates an exact match of text with more than 1K tokens – we only provide the massive evaluation of up to 512 tokens as it's substantially slower to optimize.

---

[1] https://github.com/
[2] https://civitai.com/

| Text Tokens | Prompt Tuning | | Binning Strategy | Perturbed Images | | | Img-to-Text Reconstruction | | |
|---|---|---|---|---|---|---|---|---|---|
| | Exact Match ↑ | Avg Time (s) ↓ | | △CLIP ↑ | DINO ↓ | FID ↓↑ | Hist Dist ↓ | Original Img ↑ | Rescaled Img ↑ |
| 32 | **100.0%** | **4.87** | RG Color | -2.80 | 0.0462 | 141.47 | **0.0** | **99.7%** | **97.3%** |
| | | | Random | -0.25 | 0.0010 | 9.37 | **0.0** | **99.7%** | **97.3%** |
| 64 | 99.7% | 6.19 | RG Color | -2.78 | 0.0461 | 141.29 | **0.0** | **99.7%** | 96.0% |
| | | | Random | -0.25 | 0.0010 | 9.49 | **0.0** | **99.7%** | 95.7% |
| 128 | 99.7% | 11.16 | RG Color | -2.68 | 0.0459 | 139.16 | **0.0** | 97.7% | 88.3% |
| | | | Random | -0.25 | 0.0010 | 9.38 | **0.0** | 97.7% | 91.7% |
| 256 | 99.7% | 20.06 | RG Color | -2.80 | 0.0461 | 140.24 | **0.0** | 92.0% | 79.3% |
| | | | Random | -0.26 | **0.0009** | **9.36** | **0.0** | 92.3% | 79.7% |
| 512 | 98.3% | 300.08 | RG Color | -2.87 | 0.0461 | 141.57 | **0.0** | 81.7% | 56.0% |
| | | | Random | **-0.23** | 0.0010 | 9.52 | **0.0** | 81.0% | 57.3% |

Table 1: Quantitative evaluation of OGCT via color histograms. The best results are shown in **bold**.

For the histogram-conditioned image generation, we present the results of CLIP-Score (compliance to textual prompt), DINO-Score (structure similarity), and FID (embedding similarity). Overall, we can observe the same trend as in qualitative results, where the output images with RG binning exhibit a stronger drift of metrics towards the unfavorable direction and it is likely due to the change of the controlled color channels during histogram matching. Meanwhile, the quantitative evaluation for images transformed with random binning is both visually similar and metric-wise close. For the reconstruction from image to the encoded text, we look into the pixel-level histogram distance as well as the success ratio of exact text decoding from both original and rescaled images (with a uniformly sampled factor between 0.5 and 2.0). We first observe that the color histogram of the transformed image all matches with the target one (i.e., zero distance), which credits to the constraint of our optimal transport formulation. Besides, the exact text reconstruction rate gradually decreases as we enlarge the length of target text sequence in either binning strategy and there is no significant performance gap between them. Considering the weird coloring effect that is likely to be caused by color binning, the random binning approach seems to be more favorable. In the case of target sequence with 512 tokens, the reconstruction rate is roughly 80% for the untouched image, whereas it drops to slightly about 50% in the rescaled case because of the loss of certain pixels after rescaling.

**Failure Case Analysis.** The unsuccessful text decoding can be attributed to the following causes: 1) The optimizer fails to find a soft prompt embedding that generates the exact text under the current optimization setup (e.g., learning rate scheduling, max train steps, fixed norm, etc.); 2) the rounding error introduced while mapping continuous soft prompt embedding to some discrete histogram up to the granularity of pixel numbers; 3) the loss of certain pixels after random rescaling. The first cause can be alleviated by finding optimization configurations, whereas the second and third can be a bit tricky to tackle. One potential solution is to enlarge the image resolution for better fault tolerance.

## 4 DISCUSSION

### 4.1 LIMITATIONS & FUTURE WORK

The limitation of the proposed OGCT approach mainly lies in the robustness when images get skewed. The current approach is inherently invariant to permutation but may fail to reconstruct the encoded text under some other transformations, such as non-90-degree rotations, cropping, and color jitter. We perform robustness evaluation in Sec.B. Compared to prior works that bake QR codes into images, the histogram matching approach gains better flexibility for content generation but inevitably degrades the robustness concerning the transformations mentioned above. Moreover, we may observe some slight artifacts in some output images when zooming in, which is introduced in the process of histogram matching in pixel space.

Moreover, We wish to emphasize that OGCT via color histogram is not the only solution for the perturbation function $\varphi(\cdot)$. We further extend the OGCT framework to the latent space via VQ-VAE (van den Oord et al., 2017) in Sec. C. Besides, there are far more options that are yet to be explored, such as the Fourier space and other latent space with semantic meanings.

### 4.2 BROADER IMPACT

The proposed method enables the encoding of up to thousands of text tokens within an image while maintaining a near-indistinguishable visual appearance. It has the potential to facilitate applications

in secure communication by embedding information directly into images, where the generative models serve as both the encoder and decoder. However, this technique also presents risks, particularly in its ability to conceal harmful or malicious information within innocuous-seeming images. Unlike QR codes or visible markers, which can raise suspicion or be flagged by traditional security systems, natural images encrypted via this method are less likely to be detected. This poses potential concerns where the technology could be misused to propagate illicit content or circumvent monitoring systems. To this end, we respectfully bring the readers' attention to a new field that may arise from this technique – "**generative encryption & decryption**".

Due to the inherent flexibility of large language models (LLMs) and the specialized nature of the encoder-decoder pair, decryption of the hidden information without the exact corresponding decoder model may be practically impossible. Based on our preliminary experiments, we found that fine-tuning a Llama3.1 for 10 steps with a small learning rate is sufficient to deactivate a soft prompt embedding for information decoding. Accordingly, the development of robust detection mechanisms will be critical to counter such misuse. While this encryption method offers significant advancements in secure data transmission, it equally demands ethical oversight and responsible deployment to mitigate risks associated with its potential for abuse. One potential solution is to train a classifier to detect the perturbed images as opposed to the untouched ones based on the artifact in images.

## 5 RELATED WORK

**Multimodal Representation Learning**. Multi-modal model integrates data from different modalities like images and text into a shared representation space. Models pre-pretrained using contrastive loss (van den Oord et al., 2018) have generated significant excitement as a powerful tool for data integration (Jia et al., 2021b; Li et al., 2021; Xu et al., 2021; Zhang et al., 2022). Recent researches frame the embeddings through various other techniques, including metric learning (Frome et al., 2013), multilabel classification (Joulin et al., 2015), n-gram language learning (Li et al., 2017), captioning (Desai & Johnson, 2021), and unified encoding (Girdhar et al., 2022).

**Cycle Learning**. The concept of cycle learning has been explored in various multi-modal and uni-modal contexts. Guo et al. (2020) use a cycle framework to jointly learn the transformation between text and graph. In the image domain, cycle consistent adversarial networks are used to transfer within images (Zhu et al., 2017) and between text and images (Gorti & Ma, 2018). The cycle consistency could also be applied to retrieval tasks (Cornia et al., 2018).

**Constrained Diffusion Generation**. Constrained Diffusion Generation has recently emerged as a powerful approach for generating high-quality samples under various constraints. In addition to in-painting as proposed by Rombach et al. (2022), recent techniques allows for precise control over specific attributes of the generated images (Zhang et al., 2023; Brooks et al., 2023; Tumanyan et al., 2023). Optimization-based methods have also been explored in the context of constrained diffusion. Bar-Tal et al. (2023) propose optimizing the generation path within the diffusion process.

**Steganography**. Some prior works have focused on hiding data within some medium to avoid detection. CRoSS (Yu et al., 2023) hides secret images via DDIM inversion. In contrast, most existing works focus on hiding textual information: Zhou et al. (2023b) construct a bijective mapping based on flow-based model, Peng et al. (2023) hides the message in the probability distribution along with DDPM, Su et al. (2024) modifies StyleGAN (Karras et al., 2019), and Zhou et al. (2023a) encodes secret message as object contours. OGCT enables steganography when the histogram is related to the soft prompt, and it is not restricted to a specific dataset. Moreover, our method not only intervenes in the generative process but also uses generative models for exact information decoding.

## 6 CONCLUSION

We propose Optimal Generative Cyclic Transport (OGCT) a general framework that allows exact information reconstruction during cyclic transformations across modalities (e.g., text $\rightarrow$ image $\rightarrow$ text). We exemplify the OGCT framework using a histogram-based approach, where a medium vector trained from prompt tuning can encode up to thousands of text tokens. We embed the medium vector into color or latent histograms of a visually indistinguishable image by intervening in the diffusion process and optimal transport. Such techniques empower numerous novel applications related to histogram-constrained generation, where the histogram entails the soft prompt for steganography and may connect to other visually or semantically meaningful spaces for conditional generation.

486
487
488
489
490
491
492
493
494
495
496
497
498
499
500
501
502
503
504
505
506
507
508
509
510
511
512
513
514
515
516
517
518
519
520
521
522
523
524
525
526
527
528
529
530
531
532
533
534
535
536
537
538
539

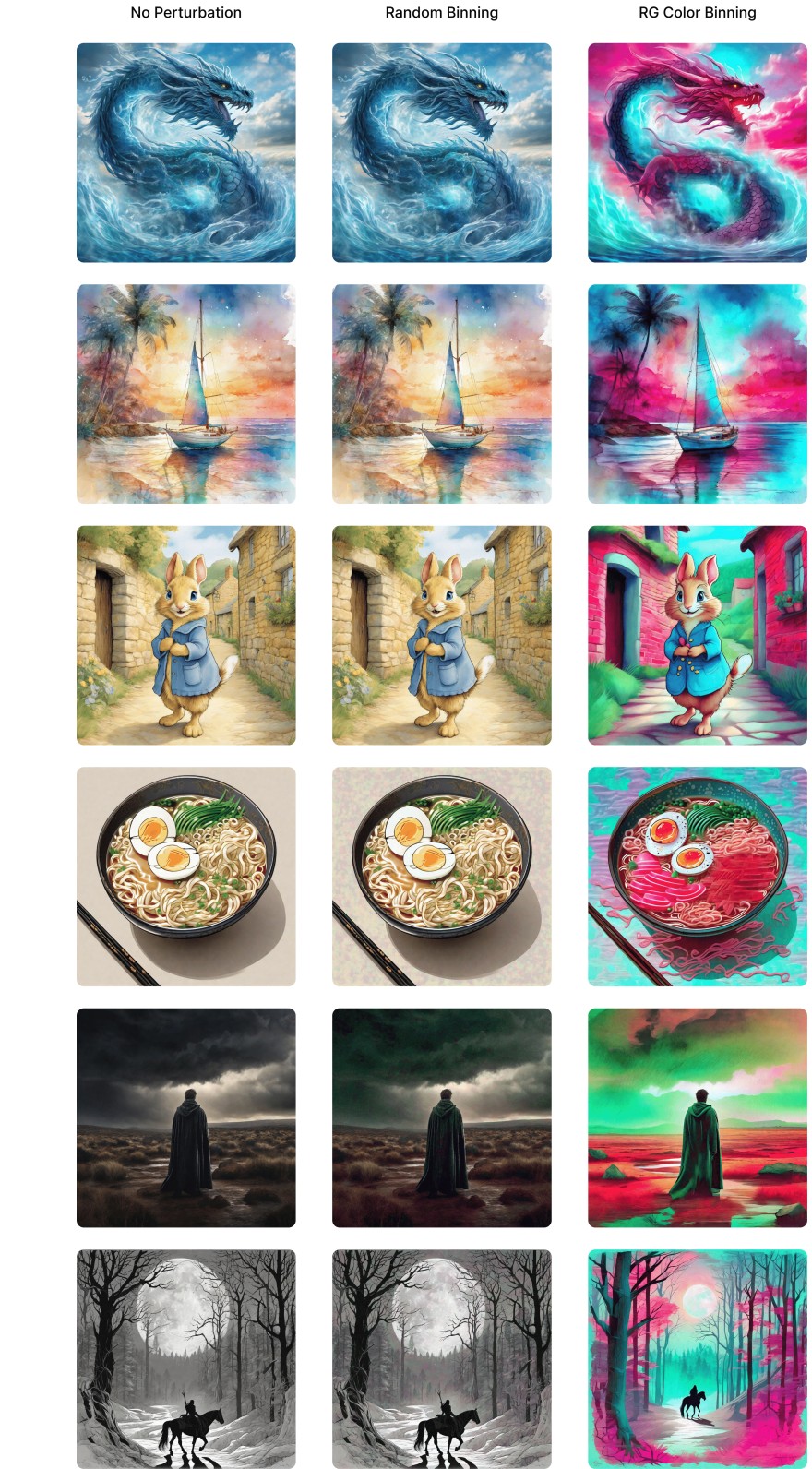

Figure 4: More qualitative results of the output images. Zoom in for the best view.

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

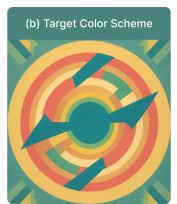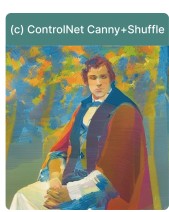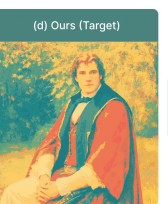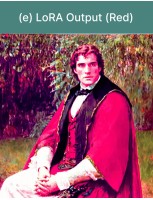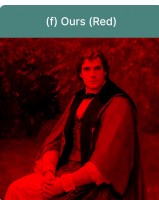

Figure 5: Recoloring comparison with other candidate methods. Better view with color.

## A    OTHER CANDIDATE APPROACHES

One may argue that existing controllable image generation techniques, such as ControlNet and Lo-RAs, can achieve precise control of color histograms. Hence, we conduct some preliminary experiments to demonstrate the insufficiency of such approaches.

ControlNet is a popular architecture to manipulate image generation models by adding additional controllable guidance, such as sketches, depth maps, poses, and so on. Among all the ControlNet variants, ControlNet-Shuffle is the closest one to our objective, which alters the spatial and color arrangements of an input image based on a target image. Fig. 5(c) presents the qualitative results of applying the ControlNet-Shuffle checkpoint (along with the Canny edge checkpoint for structure preservation) to the target image in Fig. 5(b), yet still resulting in obvious mismatch of color visually.

LoRA (Low-Rank Adaptation) is a parameter-efficient fine-tuning method that reduces the number of trainable parameters by injecting low-rank matrices into the weight matrices of a pre-trained model. Instead of updating the entire model, LoRA only modifies these additional low-rank matrices, making fine-tuning more efficient regarding memory and computation. For our case, we add an extra LoRA residual flow that takes in color histogram as extra condition and we trains it using images transformed with color jitter as targets. Fig. 5(e) shows a sample condition with a full red histogram, where the output image is still far from ideal.

In comparison, our OT-based approach in Fig. 5(d) and Fig. 5(f) consistently recolors the source image properly, demonstrating the robustness of our approach.

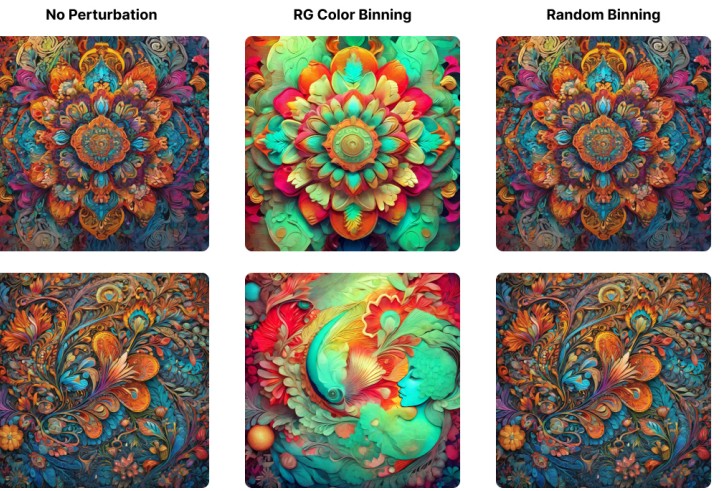

Figure 6: OGCT in pixel space with complex image and text. The output images using either binning strategy give output with proper structure and decent quality. Better view with color.

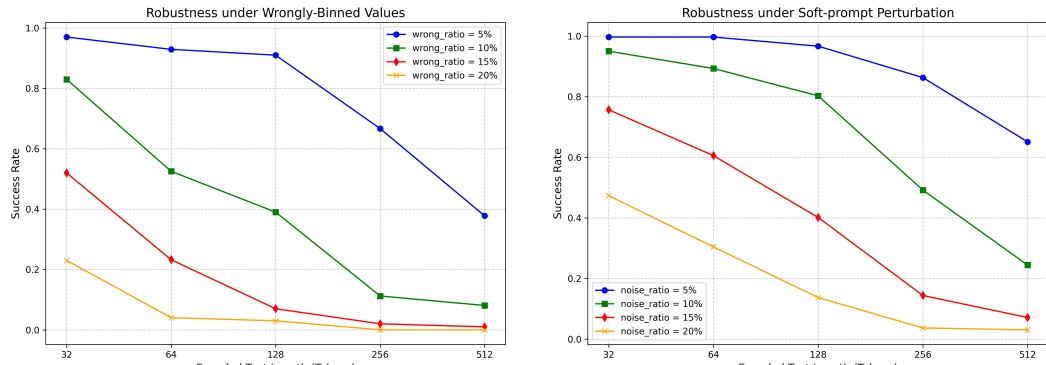

Figure 7: Robustness evaluation for histogram vectors and reconstructed soft-prompts in OGCT. For encoded text with less than 128 tokens, OGCT secures robust decoding (i.e., $> 90\%$) up to 5% of wrongly binned pixels or Gaussian noise perturbation.

## B ROBUSTNESS EVALUATION

We first evaluate the performance of OGCT in pixel space using highly complex images and text at the same time. More specifically, we generate images with intricate structures using the prompt "an artwork with intricate details, vibrant colors, high resolution, 8k" and construct strings with non-standard characters of length 128 (i.e., generally around 200 tokens) by randomly sampling from the UTF-8 encoding space. As shown in Fig. 6, both RG color binning and random color binning give image outputs with high fidelity. The success decoding rate over 300 image-text pairs is 78.7% for RG binning and 77.3% for random binning, which demonstrates the OGCT's tolerance to highly complex image and text inputs.

Then, we test the robustness of OGCT using a noisy histogram vector and reconstructed soft-prompt. For histogram vectors, we randomly distribute a subset of values to other bins and use the perturbed histogram vector for decoding; whereas for the reconstructed soft-prompts, we perform a linear interpolation with a random Gaussian vector (scaled to the same norm). The quantitative results are shown in Fig. 7. It can be seen that for text with less than 128 tokens, OGCT can still decode the text accurately (i.e., $> 90\%$ success rate) when the wrongly-binned values or the noise factor is below 5%. The performance drop becomes more significant as we enlarge the text length or increase the perturbation strength.

We note that the wrongly-binned scheme can approximate the result of common perturbations such as color jittering, blurring, and Gaussian noise, and it is applicable to all binning strategies and space of operation (e.g., pixel space and latent space). In addition, OGCT is robust to permutation (e.g., rotation) at the courtesy of histograms, but it is inevitably sensitive to cropping, as the pixels are not distributed uniformly in general.

Lastly, we empirically found that OGCT in pixel space is inherently less robust under JPEG compression. This is because: 1) the color histogram intervention and the compression operation occur both in the pixel space, which causes a conflict between the two objectives; 2) JPEG compression is conducted in the YCbCr color space, where the transformation is lossy and incurs rounding errors, resulting in the deviations of most pixel values. To this end, we extend the OGCT framework to latent space to get better tolerance to JPEG compression, while demonstrating the generalization ability of OGCT at the same time.

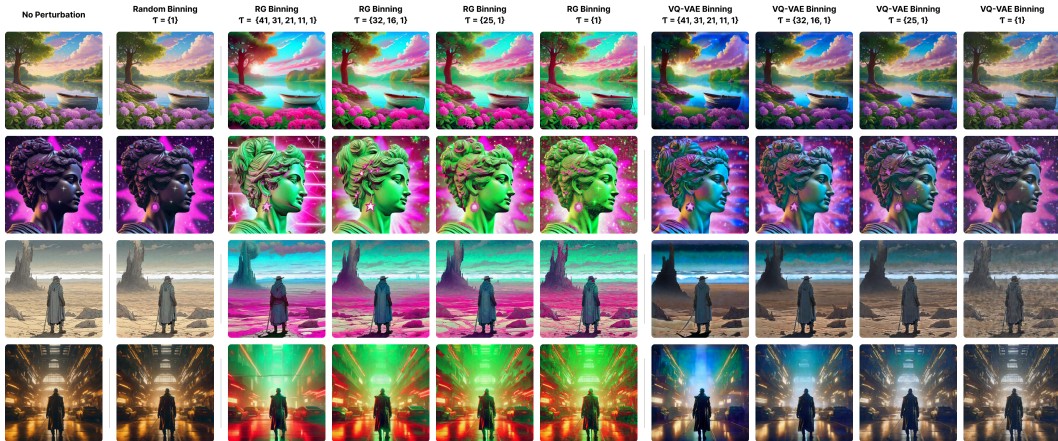

Figure 8: Qualitative results of OGCT in pixel and latent space. Better view with color.

## C  OGCT IN LATENT SPACE

We further extend the binning strategy from pixel space to latent space by taking advantage of VQ-VAE (van den Oord et al., 2017), where compact and semantically meaningful representations are learned via the quantized codebook. The general procedure of OGCT in latent space is almost the same as in pixel space, and the only difference lies in the implementation of the perturbation function $\varphi$. In general, we first decode $\mathbf{x}_0^t$ from $\mathbf{z}_0^t$ using the SDXL VAE (Podell et al., 2023) to get the predicted trajectory endpoint in pixel space. Then, we convert $\mathbf{x}_0^t$ to a quantized latent code map via a VQ-VAE, where latent histograms can be derived by counting the latent code indices. Accordingly, we can apply OT to match the latent histogram to some target distribution. Finally, we return to the SDXL latent space via VQ-VAE decoding (to $\mathbf{x}_0^{t'}$) and VAE encoding (to $\mathbf{z}_0^{t'}$).

In particular, we adopt a VQ-VAE with 4 latent channels and a codebook of size 8192. We further divide the quantized latent code into 1024 latent bins for histogram matching. The VQ-VAE is trained on the CelebA dataset of resolution $256 \times 256$, and it is sufficient to give some descent perturbation results for images of resolution $1024 \times 1024$. Some preliminary results of VQ-VAE binning and its comparison with OGCT in pixel space can be found in Fig. 8.

By operating in the latent domain, we reduce the computational complexity of OT while preserving the essential structure of the data. As each latent pixel captures higher-order features for a set of color pixels, OGCT in latent space is inherently more robust to noise and variations of the input. After incorporating DiffJPEG (Reich et al., 2024) for a pre-output optimization, we can find an exact image that matches certain latent histograms after JPEG compression and VQ-VAE encoding. Through this post hoc processing, we are essentially looking for an image that can cancel out the compression effect and give the exact histogram in the latent space. Our experiments over 20 random images resulted in a success rate of 95%.

We also wish to emphasize that the provided qualitative results aim to demo the feasibility of performing OGCT in the latent space. The adopted VQ-VAE checkpoint still has sufficient improvement possibilities, as it is fully trained on the domain of human faces. One can obtain better results by: 1) performing VQ-VAE training with higher resolutions, more diverse images, and a larger latent codebook size, which can generally lead to a more balanced color set in the latent space, and 2) baking DiffJPEG approximator into the training pipeline to make the VQ-VAE model inherently robust to JPEG compression. We leave these as directions for future work.

## D  DIFFUSION PROCESS INTERVENTION

**Choice of perturbation time steps.** We present the qualitative results of using different sets of perturbation time steps $\mathcal{T}$ in Fig.8. It can be seen that for both RG binning in pixel space and VQ-VAE binning in latent space, intervening in the diffusion process by perturbing the intermediate

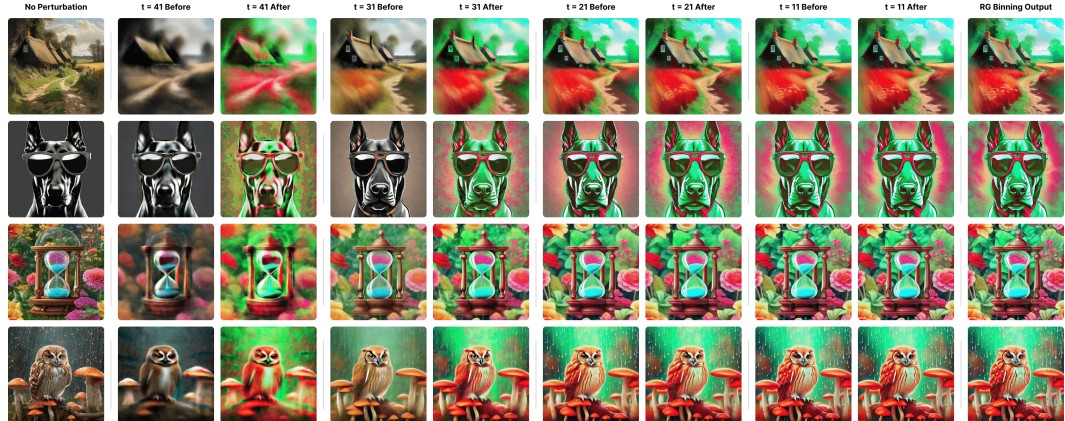

Figure 9: VAE-decoded $\mathbf{z}_0^t$ predictions using RG color binning. Better view with color.

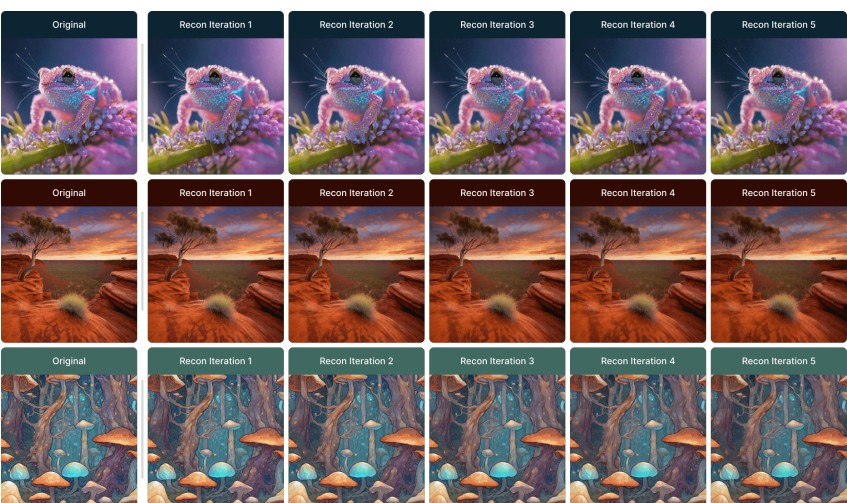

Figure 10: Content stability of SDXL VAE (Podell et al., 2023).

predictions of $\mathbf{z}_0^t$ gives smoother and more natural outputs. In comparison, simply adjusting the images before output ($\mathcal{T} = \{1\}$) may potentially result in coarse and blurry images.

**Visualization of intermediate predictions.** We showcase a few examples of VAE-decoded $\mathbf{z}_0^t$ predictions before and after RG color perturbation in Fig. 9, where we use 50 sampling steps for generation. We can observe that: 1) large pre-trained diffusion models like SDXL are capable of estimating the trajectory endpoint accurately at earlier time steps; 2) the color perturbation may lead to coarse and blurry prediction in the earlier stage, but this could be corrected and naturalized along with the diffusion process. We also demonstrate the content stability of SDXL VAE after multiple rounds of reconstruction in Fig.10, which serves as the cornerstone for diffusion process intervention.

**Implementation details for `OT-histogram-matching`.** The procedures for histogram matching are as follows: 1) divide the input pixel or latent map into a set of bins based on the selected binning strategy; 2) calculate the target histogram from the soft-prompt embedding and round to a vector of integers; 3) calculate the cost matrix between each source and target bin (typically L2 distance between vectors); 4) solve for the optimal transport plan using `ot.emd` from the Python Optimal Transport library[3]; 5) for each pair of source and target bins, randomly sample pixels from the source bin based on the optimal transport plan, and set them to the corresponding values of the target bin. The source code can be found in the supplementary material.

---

[3]https://pythonot.github.io/

