# OpenReview forum: "Optimal Generative Cyclic Transport between Image and Text"
_ICLR.cc/2025/Conference — Submitted to ICLR 2025_

### Official Review · Reviewer_dAGu · 2024-11-02

**Soundness:** 3
**Presentation:** 3
**Contribution:** 3
**Rating:** 6
**Confidence:** 3

**Summary:**

This work proposes a novel text-image-text reconstruction method in cross-modality generation by guiding generative models with a deterministic function. The authors use color histograms as constraints to perform precise reconstruction, and this method can be applied for data protection.

**Strengths:**

1. The authors propose a novel and effective method for the cyclic transformation of text → image → text.

2. The manuscript is well-written and easy to understand.

3. Both the visualization and quantitative results seem promising.

4. The use of color histogram guidance is interesting.

**Weaknesses:**

1. Additional robustness analysis is needed to evaluate the effects of noise, blur, JPEG compression, and other transmission channel distortions.

2. The first section should ideally begin with a discussion on common applications of text → image → text translation.

3. There are already numerous generative methods for data protection, so the contribution stated in line 442 seems somewhat overstated. The authors should discuss their differences from existing methods in more detail, such as:

    [r1] StegaStyleGAN: Towards Generic and Practical Generative Image Steganography (AAAI 24)

    [r2] Cross: Diffusion Model Makes Controllable, Robust, and Secure Image Steganography (NeurIPS 24)

    [r3] StegaDDPM: Generative Image Steganography based on Denoising Diffusion Probabilistic Model (MM 23)

    [r4] Generative Steganography via Auto-Generation of Semantic Object Contours (TIFS 24)

    [r5] Secret-to-Image Reversible Transformation for Generative Steganography (TDSC 23)

4. There are a few typo errors throughout the manuscript. For example, in line 391, “a100” should be corrected to “A100.”

**Questions:**

Please refer to Weaknesses.

---

> ### Author Response · Authors · 2024-11-25
>
> Dear Reviewer dAGu,
>
> Thank you for taking the time to review our submission. We appreciate your thoughtful and detailed feedback and your effort in engaging with our work. We have added more experiments requested by the reviewers to the Appendix. We summarize the answers to your concerns as follows:
>
> **Q: OGCT's robustness under perturbation and compression**
>
> We test the robustness of OGCT using wrongly binned histograms and noisy soft prompts. For histogram vectors, we randomly distribute a subset of values to other bins and use the perturbed histogram vector for decoding; for the soft prompts, we perform a linear interpolation with a random Gaussian vector (scaled to the same norm). The quantitative results indicate that for text with less than 128 tokens, OGCT can still decode the text accurately (i.e., $>90\%$ success rate) when the wrongly-binned values or the noise factor is below 5\%. **(More details in Appendix Sec.B \& Fig.7)**
>
> We empirically found that OGCT in pixel space is inherently less robust under JPEG compression due to the conflict between histogram intervention and data compression. To this end, we extend the OGCT framework to latent space to get a better tolerance to JPEG compression and demonstrate the generalization ability of OGCT.
>
> For OGCT in latent space, we trained a VQ-VAE to encode images to quantized latent vectors. Then, we apply histogram binning and matching using quantized codebook indices. By operating in the latent domain, we reduce the computational complexity of OT and preserve the essential structure of the data. After incorporating DiffJPEG [r6] for a pre-output optimization, we can find an exact image that matches certain latent histograms after JPEG compression and VQ-VAE encoding. Our experiments over 20 random images resulted in a success rate of 95\%. **(More details in Appendix Sec.C \& Fig.8)**
>
> **Q: Difference with related works on steganography**
>
> Many thanks for providing the related works in detail, we further summarize the key problem/difference as follows:
>
> [r1][r4][r5] require dataset-specific training, whereas our approach applies to any input.
>
> [r2] adopts a different task formulation, where they aim to hide images, using text/model checkpoint as the key.
>
> [r3] does not consider the aspect of robustness in their paper, but its approach should be sensitive to most perturbations. In contrast, our approach still exhibits tolerance to noisy input to some extent.
>
> **Q: Fixing write-up**
>
> The typo has been fixed -- thanks for pointing that out. The introduction is subject to revision, as we have extended the proposed framework to latent space binning and histogram-constrained image generation in general. Most new results have been tentatively placed in the Appendix.
>
> Please let us know if you have any further questions.
>
> [r6]: Differentiable JPEG: The Devil is in the Details (WACV 2024)

---

### Official Review · Reviewer_fK7m · 2024-11-05

**Soundness:** 2
**Presentation:** 2
**Contribution:** 2
**Rating:** 5
**Confidence:** 5

**Summary:**

This paper propose a method to generate images more aligned with the text prompt using optimal generative cycle consistency between image and text. They ensure the image consistency, using color histogram matching using optimal transport and show the generated images are natural looking for various tasks.

**Strengths:**

1. The idea of using optimal transport for color histogram matching is interesting.

2. The generated results looks good, to some extent.

**Weaknesses:**

1. Why matching color histogram is enough to generate consistent images? E.g., it might happen that two images have exact same color histogram, but totally differnet content. Color domain discripency makes sense if the diffusion model is conditioned on images, but for text-to-image diffusion model, it is highly unlikely that the prompt can capture the exact details. Please clarify this.

2. There are works which address the issue of (image->text->image) consitency using pretrained caption and diffusion models, comparing with those methods need to be done.

3. Exact prompt and seed might vary across machines, implementations, how the authors justify this? E.g., they attempt to retreive the specific diffusion model and seeds from the images, which I am not convinced to be extracted using these details. Please clarify.

4. Instead of just visualization, the authors might quantify their approach for some recognition tasks, e.g., captioning, classififcation etc.

5. The paper lacks coherence sometimes, e.g., in the abstract they mentioned about steganography, but it is not explored anywhere.

6. Overall writing and flow need to be improved.

[1] Roy et al. "Cap2aug: Caption guided image to image data augmentation". WACV 2025

[2] Kondapaneni et al. "Text-image Alignment for Diffusion-based Perception", CVPR 2024

**Questions:**

Please justify the weakness.

---

> ### Author Response · Authors · 2024-11-25
>
> Dear Reviewer fK7m,
>
> Thank you for taking the time to review our submission. We appreciate your effort in engaging with our work. We believe there may have been some misinterpretation of the task and method proposed in our paper. To clarify and address this, we would like to **restate our task and methodology** in more detail below.
>
> OGCT originates from the idea of achieving exact reconstruction for text $\rightarrow$ image $\rightarrow$ text transformation. In other words, after text-to-image generation using diffusion models, we wish to decode the original text from the generated image.
>
> We divide this task into two sub-problems: 1) intervene in the diffusion generation process so that the output image entails a medium vector (e.g., color histogram in pixel space, latent code histogram in VQ-VAE space, etc.); 2) find a way to decode exact information from this medium vector.
>
> We tackle the first sub-problem by adjusting the diffusion trajectory endpoint during generation, where the adjustment is based on a solved optimal transport (OT) plan that matches the source input to some target distribution. We tackle the second sub-problem using soft prompt tuning -- basically tuning a single trainable token in the LLM embedding space that can condition the model to generate the exact encoded text. On top of that, we design some deterministic functions to handle the conversion between embedding vectors and histograms.
>
> We also note that the aforementioned framework can be further generalized in the following ways: 1) the encoded text can be irrelevant to the generated images, so we can hide arbitrary forms of text inside the image; 2) the hidden medium vector is not restricted to soft prompt, as the proposed framework is designed for histogram-constrained generation with fine-grained control, such as generating images with some exact-form histogram (e.g., color, frequency, visually or semantically meaningful tokens in latent space, etc.).
>
> We summarize the answers to your concerns as follows:
>
> **Q: Why matching color histogram is enough to generate consistent images?**
>
> Our objective is to intervene in the diffusion process so that the output image matches a certain histogram. In this way, we can decode exact information from the output image using the LLM. Accordingly, for images that share the same histogram, the content could be different, but the decoded information will be the same (when using the same binning strategy).
>
> **Q: Comparison with image $\rightarrow$ text $\rightarrow$ image works**
>
> The initial objective of OGCT is to tackle the cyclic transformation text $\rightarrow$ image $\rightarrow$ text, which is not directly related to the suggested task. Please let us know if there are closely related works in this field.
>
> **Q: Exact prompt and seed might vary across machines, implementations.**
>
> Given the same binning strategy, we can obtain the same histogram vector. After reconstructing it to the soft prompt embedding via deterministic functions, we can use it to condition the pre-specified LLM to generate exact-match sentences, regardless of machines and random seeds.
>
> **Q: Comparison with classic recognition tasks**
>
> Our framework is inherently different from classic recognition tasks, such as captioning and classification. In those tasks, the ground-truth labels are directly related to the image content; whereas in our case, the encoded information is not necessarily related to the image content, and the key objective is to decode the hidden medium vector (i.e., some sort of histogram).
>
> Please let us know if you have any further questions.

---

> > ### Comment · Reviewer_fK7m · 2024-11-27
> > **Response to authors**
> >
> > Thanks for the rebuttal, but my concerns are not well addressed. E.g., if the content is different, but the color histogram is same, how the decoded information is same? In that case the essence of generative models are not used. Also, I believe that comparing with the above-mentioned works needs to be done for generalization. If the authors ignore that and claim this only  deals with steganography, there’s also a vast literature for that. Overall, I’m keeping my rating.

---

> > > ### Author Response · Authors · 2024-11-28
> > >
> > > Dear Reviewer fK7m,
> > >
> > > Thank you for the reply. It seems that we may not have fully understood the question you raised (i.e., ``E.g., if the content is different, but the color histogram is same, how the decoded information is same?"). To clarify, we hypothesize that you might perceive our work as aiming to better align the generated image and the textual information by utilizing the histograms. However, that is not our focus. Our objective is to ensure that the generated image adheres to a user-defined constraint in terms of a deterministically computable histogram. If the computable histogram encapsulates semantic information, then the hypothesized perspective can be valid.
> > >
> > > In our work, we showcase two examples of histogram constraints: (1) color histograms, which do not contain semantic information but ensure that the generated image's color distribution meets the specified requirement, and (2) a histogram derived from a VQ-VAE. This demonstrates that any histogram computable by a model (e.g., latent representations) can serve as a constraint. If the VQ-VAE histogram carries semantic information, the generated images will reflect that information accordingly.
> > >
> > > In experiments, we demonstrate that our approach can be applied with constraints with arbitrary values (a soft prompt for LLMs). This particular case ensures the cyclic generation of text $\Rightarrow$ image $\Rightarrow$ text and illustrates the broad applicability and robustness of our method.
> > >
> > > Lastly, we address the differences with the aforementioned works. [1] is a caption-guided data augmentation technique, where they pair an actual image with the caption of another image to generate novel data points using diffusion-based image-to-image translation models. [2] leverages image captioning to inject additional information into diffusion-based predictors via the cross-attention layers, which enhances performance in classic image recognition tasks (e.g., depth estimation, segmentation, object detection). In contrast, our work is about constrained generation. We compared VLM image captioning with a special case of OGCT in Fig. 1, where the constraint was the histogram derived from a soft prompt. This comparison aimed to illustrate that our framework enables exact information decoding during the text $\Rightarrow$ image $\Rightarrow$ text transformation. Beyond this, our method allows the use of constraints that can be entirely unrelated to the image content. We will revise this section in the next version of our paper to avoid potential confusion.
> > >
> > > We hope this could address your concerns to some extent, and we would greatly appreciate it if you could further elaborate on your question.

---

### Official Review · Reviewer_G8LH · 2024-11-06

**Soundness:** 3
**Presentation:** 3
**Contribution:** 3
**Rating:** 6
**Confidence:** 3

**Summary:**

The paper presents a novel approach, Optimal Generative Cyclic Transport (OGCT), aimed at achieving exact, cyclic transformations between text and image modalities. The core of this work involves encoding text into images using a soft embedding that aligns with a color histogram, which can be recovered deterministically. The technique leverages diffusion models and optimal transport to generate images that encapsulate the encoded text as a recoverable soft prompt. The paper proposes and evaluates multiple binning strategies, with the random binning having perceptually indistinguishable outputs from the unencoded images

**Strengths:**

- The paper addresses an important challenge in multimodal generative models, where information is typically lost during cyclic transformations. This application could be instrumental for secure communication and data integrity.

- The introduction of a reversible soft embedding that maps text to color histograms, recoverable through a deterministic optimal transport algorithm, represents a novel approach in the field.

- The method is described in a structured manner, with an explicit algorithm provided for the entire OGCT process. This detailed exposition facilitates understanding and reproducibility.

- The random binning algorithm effectively enables the storage of a large quantity of text information within images while keeping the visual impact minimal, showing potential in practical scenarios.

**Weaknesses:**

- The method's reliance on color histograms raises concerns about robustness. While the authors have demonstrated resilience to rescaling, the algorithm may be sensitive to other common augmentations like color jitter, cropping, rotation, blur, and Gaussian noise. Testing against these transformations is necessary to confirm its applicability in realistic, potentially hostile scenarios.

- The experiments presented do not cover a comprehensive analysis of the algorithm. Including experiments such as variable perturbation time steps and perturbation strengths would provide deeper insights into the algorithm’s robustness and limitations.

**Questions:**

- In Section 2.4, the authors introduce the $\varphi$ function for histogram matching in the diffusion model but do not provide clarity on its calculation through the optimal transport approach. It would be beneficial if the authors included pseudo-code or an explanation of how the "OT-histogram-matching" function is implemented.

- Could the authors clarify the selection of perturbation steps, $\tau$, used in the experiments? Additionally, has there been any investigation into how using fewer or different sets of perturbation steps impacts the performance, especially if perturbations are applied only at the last generation step ($t=1$)?

- If the method is inherently unsuited to handle augmentations such as color jitter, what alternative solutions do the authors suggest? Could other functions besides color histograms be incorporated into the algorithm? The paper would gain greater value by incorporating a more generalized set of transformations alongside color histograms to enhance the robustness of the method.

- The suggested algorithm performs VAE-decoding on $z_0^t$ values calculated during the generation process. Can the authors provide a study or an experiment they performed to show the reliability of using VAE-decoding on these approximate values of $z_0$?

---

> ### Author Response · Authors · 2024-11-25
>
> Dear Reviewer G8LH,
>
> Thank you for taking the time to review our submission. We appreciate your thoughtful and detailed feedback and your effort in engaging with our work. We have added more experiments requested by the reviewers to the Appendix. We summarize the answers to your concerns as follows:
>
> **Q: OGCT's robustness under perturbation and compression**
>
> We test the robustness of OGCT using wrongly binned histograms and noisy soft prompts. For histogram vectors, we randomly distribute a subset of values to other bins and use the perturbed histogram vector for decoding; for the soft prompts, we perform a linear interpolation with a random Gaussian vector (scaled to the same norm). The quantitative results indicate that for text with less than 128 tokens, OGCT can still decode the text accurately (i.e., $>90\%$ success rate) when the wrongly-binned values or the noise factor is below 5\%. **(More details in Appendix Sec.B \& Fig.7)**
>
> We empirically found that OGCT in pixel space is inherently less robust under JPEG compression due to the conflict between histogram intervention and data compression. To this end, we extend the OGCT framework to latent space to get a better tolerance to JPEG compression and demonstrate the generalization ability of OGCT.
>
> For OGCT in latent space, we trained a VQ-VAE to encode images to quantized latent vectors. Then, we apply histogram binning and matching using quantized codebook indices. By operating in the latent domain, we reduce the computational complexity of OT and preserve the essential structure of the data. After incorporating DiffJPEG [r1] for a pre-output optimization, we can find an exact image that matches certain latent histograms after JPEG compression and VQ-VAE encoding. Our experiments over 20 random images resulted in a success rate of 95\%. **(More details in Appendix Sec.C \& Fig.8)**
>
> **Q: Implementation of OT-histogram-matching**
>
> The procedures for histogram matching are as follows: 1) divide the input pixel or latent map into a set of bins based on the selected binning strategy; 2) calculate the target histogram from the soft prompt embedding and round to a vector of integers; 3) calculate the cost matrix between each source and target bin (typically L2 distance between vectors); 4) solve for the optimal transport plan using \texttt{ot.emd} from the Python Optimal Transport library; 5) for each pair of source and target bins, randomly sample pixels from the source bin based on the optimal transport plan, and set them to the corresponding values of the target bin. The source code can be found in the supplementary material. We will add pseudo-code in the next version of the paper.
>
> **Q: Choice of perturbation time steps**
>
> In general, intervening in the diffusion process by perturbing the intermediate predictions of $\textbf{z}_0^t$ gives smoother and more natural outputs. In comparison, simply adjusting the images before output ($\mathcal{T} = \{1\}$) may potentially result in coarse and blurry images. **(More details in Appendix Sec.D \& Fig.8)**
>
> **Q: Reliability of $\textbf{z}_0^t$ prediction and SDXL VAE**
>
> We showcase a few examples of VAE-decoded $\textbf{z}_0^t$ predictions before and after RG color perturbation in Fig.9. We can observe that: 1) large pre-trained diffusion models like SDXL are capable of estimating the trajectory endpoint accurately at earlier time steps; 2) the color perturbation may lead to coarse and blurry prediction in earlier stage, but this could be fixed along with the diffusion process. We also demonstrate the content stability of SDXL VAE after multiple rounds of reconstruction in Fig.10. **(More details in Appendix Sec.D)**
>
> Please let us know if you have any further questions.
>
> [r1]: Differentiable JPEG: The Devil is in the Details (WACV 2024)

---

### Official Review · Reviewer_Vkg7 · 2024-11-08

**Soundness:** 4
**Presentation:** 4
**Contribution:** 3
**Rating:** 5
**Confidence:** 5

**Summary:**

The paper presents the Optimal Generative Cyclic Transport (OGCT) framework, aimed at achieving precise, lossless cyclic transformations across image and text modalities. Leveraging color histograms as a guidance vector, the proposed framework optimizes "soft prompts" within a language model and constrains the diffusion process in a generative model to encode text-based information directly into image color histograms. The encoded images are then decodable to the original text sequence using the same histogram vector, making this framework robust and flexible for applications in secure communication and generative encryption.

**Strengths:**

The paper is well-organized and detailed.

**Weaknesses:**

1. The paper lacks an evaluation of OGCT’s robustness against common image compression and information degradation techniques, as demonstrated in Table 2 of the paper [1]. In real-world scenarios, images often undergo transformations that introduce minor information loss, such as JPEG compression, added noise, and other alterations, which could potentially affect the integrity of OGCT’s color histogram-based decoding. Without experiments that test OGCT’s performance under these conditions, it is unclear how well the framework would perform in practical applications where images are subject to compression or other modifications.

2. The robustness of the OGCT under different types of input text or image complexity is not thoroughly tested. The framework may encounter difficulties when encoding highly complex images or diverse language styles, particularly with non-standard characters, which could affect accuracy in real-world scenarios.

[1] Yu, Jiwen, et al. "Cross: Diffusion model makes controllable, robust and secure image steganography." Advances in Neural Information Processing Systems 36 (2024).

**Questions:**

See weakness.

---

> ### Author Response · Authors · 2024-11-25
>
> Dear Reviewer Vkg7,
>
> Thank you for taking the time to review our submission. We appreciate your thoughtful feedback and your effort in engaging with our work. We have added more experiments requested by the reviewers to the Appendix. We summarize the answers to your concerns as follows:
>
> **Q: OGCT's robustness under perturbation and compression**
>
> We test the robustness of OGCT using wrongly binned histograms and noisy soft prompts. For histogram vectors, we randomly distribute a subset of values to other bins and use the perturbed histogram vector for decoding; for the soft prompts, we perform a linear interpolation with a random Gaussian vector (scaled to the same norm). The quantitative results indicate that for text with less than 128 tokens, OGCT can still decode the text accurately (i.e., $>90\%$ success rate) when the wrongly-binned values or the noise factor is below 5\%. **(More details in Appendix Sec.B \& Fig.7)**
>
> We empirically found that OGCT in pixel space is inherently less robust under JPEG compression due to the conflict between histogram intervention and data compression. To this end, we extend the OGCT framework to latent space to get a better tolerance to JPEG compression and demonstrate the generalization ability of OGCT.
>
> For OGCT in latent space, we trained a VQ-VAE to encode images to quantized latent vectors. Then, we apply histogram binning and matching using quantized codebook indices. By operating in the latent domain, we reduce the computational complexity of OT and preserve the essential structure of the data. After incorporating DiffJPEG [r1] for a pre-output optimization, we can find an exact image that matches certain latent histograms after JPEG compression and VQ-VAE encoding. Our experiments over 20 random images resulted in a success rate of 95\%. **(More details in Appendix Sec.C \& Fig.8)**
>
> **Q: OGCT's robustness under highly complex image and text**
>
> We performed a robustness evaluation under the suggested scenario. For the image side, we generate images with intricate details and vibrant colors; for the text side, we randomly sample complicated non-standard characters from the UTF-8 encoding space (of length 128), making it highly unpredictable. The success decoding rate over 300 image-text pairs is 78.7\% for RG binning and 77.3\% for random binning, and the output images are decent in quality. In addition, enlarging the number of optimization steps for soft prompt tuning may further increase the reconstruction performance because of more accurate soft prompts). **(More details in Appendix Sec.B \& Fig.6)**
>
> Please let us know if you have any further questions.
>
> [r1]: Differentiable JPEG: The Devil is in the Details (WACV 2024)

---

### Author Response · Authors · 2024-11-25
**Overall Comments by Authors**

**We thank all reviewers for their invaluable feedback and have revised the paper and appendix accordingly. The revised parts in the original text and the newly added sections in the Appendix have been noted in blue.**

The major revisions include:

[1] The big picture about the proposed framework for histogram-constrained image generation (Sec.1, Sec.4.1, Sec.6).

[2] Related work discussion on Steganography and its relation to the proposed framework (Sec. 5).

[3] Robustness evaluation of OGCT against complex data, wrong histogram, and Gaussian noise. (Sec. B)

[4] Qualitative results for OGCT in latent space as well as its robustness against JPEG compression. (Sec. C)

[5] Further details about diffusion process intervention: choice of perturbation time steps, visualization of intermediate predictions, and histogram matching implementation. (Sec. D)

Meanwhile, we wish to further clarify a common question on the relationship between steganography and our framework:

1) **We position our framework to be a histogram-constrained generation approach, where steganography is one of its prominent applications (done by relating the histogram to the soft prompt).**

2) Unlike many existing steganography approaches that are restricted to dataset-specific scenarios, the steganography enabled by our approach is widely applicable to any text-to-image model and any form of text in a plug-and-play manner.

3) Typical steganography works tend to hide information by encoding bits or encrypted/transformed bits, whereas we are the first approach that encodes the information using an off-the-shelf generative model.

---

### Meta-Review · Area_Chair_AHd8 · 2024-12-16

**Metareview:**

In this work, the authors presents the Optimal Generative Cyclic Transport (OGCT) framework to address the issue of securing an exact match between the original and reconstructed contents for the cyclic transformation of text -> image -> text.
Overall, its main weakness is insufficient evaluations, as commonly mentioned by reviewers as follows:
Reviewer G8LH
``...I meant assessing performance when augmentations like color jitter, blurring, or Gaussian noise are applied to the images in pixel-space. The paper lacks sufficient experiments on this aspect, and the evaluation remains incomplete...including robustness tests, especially since pixel-space augmentations may affect the method's reliance on color histograms.''

Reviewer fK7m
``... concerns are not well addressed. E.g., if the content is different, but the color histogram is same, how the decoded information is same? In that case the essence of generative models are not used. Also, I believe that comparing with the above-mentioned works needs to be done for generalization...''

Reviewer dAGu
``... agree with the weaknesses pointed out by Reviewer fK7m and I believe fK7m's concerns are not well addressed....''

**Additional Comments On Reviewer Discussion:**

``Insufficient evaluation'' is the key mismatch between the authors and most reviewers.
During the rebuttal process, the authors' responses still did not satisfy all reviewers.

---

### Decision · Program_Chairs · 2025-01-22

Reject